# Unraveling Tumor Heterogeneity by Using DNA Barcoding Technologies to Develop Personalized Treatment Strategies in Advanced-Stage PDAC

**DOI:** 10.3390/cancers13164187

**Published:** 2021-08-20

**Authors:** Philip Dujardin, Anna K. Baginska, Sebastian Urban, Barbara M. Grüner

**Affiliations:** 1West German Cancer Center, Department of Medical Oncology, University Hospital Essen at the University Duisburg-Essen, 45147 Essen, Germany; anna.baginska@uk-essen.de (A.K.B.); sebastian.urban@uk-essen.de (S.U.); 2German Cancer Consortium (DKTK) Partner Site Essen/Düsseldorf, 45147 Essen, Germany

**Keywords:** PDAC, molecular barcoding, tumor heterogeneity, metastasis, therapy resistance

## Abstract

**Simple Summary:**

Pancreatic cancer is one of the hardest-to-treat cancers. This is mainly due to its heterogeneity, where subsets of cancer cells possess distinct properties and abilities that determine if and how they metastasize or respond to therapy. DNA barcoding technologies have emerged as a powerful tool to study this heterogeneity, as they allow labeling of individual tumor cells within a cancer cell pool and follow their cellular states and fates during metastasis or upon therapy. The aim of this review was to provide an overview of the various levels of tumor heterogeneity in pancreatic cancer, the obstacles these levels of heterogeneity can cause for effective personalized treatment strategies, and how different barcoding approaches can be applied to study these important questions.

**Abstract:**

Tumor heterogeneity is a hallmark of many solid tumors, including pancreatic ductal adenocarcinoma (PDAC), and an inherent consequence of the clonal evolution of cancers. As such, it is considered the underlying concept of many characteristics of the disease, including the ability to metastasize, adapt to different microenvironments, and to develop therapy resistance. Undoubtedly, the high mortality of PDAC can be attributed to a high extent to these properties. Despite its apparent importance, studying tumor heterogeneity has been a challenging task, mainly due to its complexity and lack of appropriate methods. However, in recent years molecular DNA barcoding has emerged as a sophisticated tool that allows mapping of individual cells or subpopulations in a cell pool to study heterogeneity and thus devise new personalized treatment strategies. In this review, we provide an overview of genetic and non-genetic inter- and intra-tumor heterogeneity and its impact on (personalized) treatment strategies in PDAC and address how DNA barcoding technologies work and can be applied to study this clinically highly relevant question.

## 1. Introduction

Even though the 5-year survival rate of pancreatic ductal adenocarcinoma (PDAC) has recently reached double-digit numbers at 10.8% [1], it still remains undoubtedly one of the deadliest human malignancies with an extremely poor prognosis. Among the main reasons that make PDAC one of the hardest-to-treat cancers are late diagnosis in advanced stages, high metastatic capacity, intrinsic therapy resistance, and dense and immunosuppressive desmoplastic stroma as well as multi-level heterogeneity, not only on a genetic, but also on transcriptomic, epigenetic, and metabolic levels [2,3,4]. Cancer development and evolution are dynamic events in which subsets of tumor cells gain the ability to progress, metastasize, and become resistant to therapies. As early detection in PDAC is rare due to indolent disease courses, most of the patients are not eligible for a surgical resection. Therapy of unresectable, advanced-stage metastatic PDAC is extremely challenging, characterized by a high refractoriness to currently available therapies and either de novo or acquired development of resistance [5,6]. Recently, Conroy et al. demonstrated the benefits of modified 5-fluorouracil, leucovorin, irinotecan, and oxaliplatin (mFOLFIRINOX) over gemcitabine alone, which marked a new milestone in adjuvant chemotherapeutic treatment of pancreatic cancer [7]. Yet, apart from the EGFR (epidermal growth factor receptor) inhibitor erlotinib [8], targeted therapeutic approaches have not resulted in a significant improvement of disease outcome in the last decades. The fact that only a subset of patients who develop a rash as a side effect actually benefit from erlotinib treatment already emphasizes an underlying heterogeneity in PDAC biology, indicating an urgent need for a more individualized treatment structure and stratifying biomarkers. However, this requires an in-depth understanding of the (sub-)clonal dynamics and the evolution of cellular populations that give rise to all levels of tumor heterogeneity. Here, powerful DNA barcoding technology has developed and contributed tremendously to our understanding of tumor heterogeneity on multiple levels.

## 2. Tumor Heterogeneity and Its Impact on Personalized Treatment Strategies in PDAC

On a cellular level, tumor heterogeneity is defined by phenotypical and morphological differences between tumor cells, resulting in different gene expression or epigenetic profiles, different metabolic dependencies, and distinct rates of proliferation or metastatic potential. Inter-tumor heterogeneity describes diversity between tumors or tumor sites, while intra-tumor heterogeneity refers to differences within a single tumor (site) itself (Figure 1). Tumor heterogeneity can be also observed between different sub-regions of the tumor or the tumor microenvironment (TME), which is of particular interest in digital pathology [9].

### 2.1. Inter-Tumor Heterogeneity

PDAC is a unique type of tumor, characterized by its high heterogeneity and a very rich desmoplastic stroma, which can make up to 90% of the tumor volume. Single cell RNA sequencing from PDAC patients’ biopsies showed high inter-tumor heterogeneity between patients within the cancer cells, with stromal cells being more homogenous [10]. This type of heterogeneity is mostly recognized as an underlying cause for symptoms in different patients, predisposition to early metastasis, and sensitivity to treatment.

#### 2.1.1. Genomic Alterations and Subtypes

Multiple reviews have provided great overviews on therapeutic challenges and opportunities of targeting signaling pathway-related oncogenes in pancreatic cancer [11,12,13]. On a genomic level, PDAC is quite homogenous between patients. In most cases, there is only a handful of driver mutation genes, including *KRAS* (mutated in over 90% of cases), *TP53* (60–70%), *CDKN2A* (40–50%), and *SMAD4* (30–40%) [2,14,15,16]. Whole exome or genome sequencing did not reveal any new key driver genes with prevalence higher than 20% [2,15,16]. Only a small subset of patients shows mutations in targetable oncogenes (e.g., *ERBB2*) or DNA damage repair genes, in particular in mismatch repair regulators or genes like *BRCA1*, *BRCA2,* and *PALB2* [15,17]. Occurrence of a mutation, however, does not always predict dependency and “actionability” [18,19]. In PDAC, tumor progression, metastasis, and therapy resistance seem not to be completely relying on genetic mutations, but rather on different mechanisms such as interaction with tumor microenvironment (TME), epithelial-to-mesenchymal transition (EMT), epigenetic modifications, or changes in their metabolism [20,21,22,23,24].

Even though none of the frequently mutated driver genes is correlated with a distinctive subtype, Waddell et al. were able to classify PDAC into four subtypes based on structural variation of chromosomes. These are the stable subtype (in 20% of samples) with less than 50 structural variations, the locally rearranged subtype (30%) with significant focal events located on one or two chromosomes, the scattered subtype (36%) with moderate range of nonrandom chromosomal damage, and the unstable subtype (14%) with a large number of structural variation events [15].

#### 2.1.2. Transcriptomic Subtypes

Collisson et al. defined three PDAC subtypes based on transcriptional profiles: quasi-mesenchymal (30% prevalence), exocrine-like (35%), and classical (35%) [25]. Bailey et al. proposed an overlapping classification into four groups: squamous (similar to quasi-mesenchymal), aberrantly differentiated endocrine exocrine (ADEX; exocrine-like), pancreatic progenitor (classical), and immunogenic (classical) [2]. A simpler separation into two subtypes was suggested by Moffit et al.: basal-like and classical [26]. According to a TCGA study, squamous samples showed a significant overlap with basal-like samples, whereas classical samples from Moffit et al. and Collisson et al. largely overlapped with progenitor samples from Bailey et al. [27]. In general, the quasi-mesenchymal/squamous/basal-like subtype demonstrates worse prognosis and is more aggressive, due to its resistance to chemotherapy and a more mesenchymal gene expression. The progenitor or classical subtype, on the other hand, is associated with epithelial gene expression and responds better to chemotherapy. The ADEX or exocrine-like subtype is enriched for gene programs in endocrine and exocrine development, while the immunogenic subtype shows enrichment of pathways involved in immune cell infiltration and immune signaling [2]. These last two subtypes, however, display a high similarity to the classical subtype, which might imply that they could be assigned as a subtype of the classical/progenitor group. Moreover, the TCGA study indicated a strong association of these subtypes with low purity samples, suggesting contamination from other cell types [27]. Importantly, however, Porter et al. recently demonstrated that the epithelial and quasi-mesenchymal subtypes are inducible by different therapies, further contributing to their heterogeneity, plasticity, and, therefore, resistance to therapy [28].

The transcriptomic landscape of PDAC is not completely controlled by genomic changes, but rather by epigenetic modifications—a system that is highly dysregulated in PDAC [4,29,30,31]. The transcriptomic PDAC subtypes described above can be explained in part by the epigenetic landscape, in particular, DNA methylation [29]. Unlike genetic modifications, epigenetic changes and, therefore, transcriptomic programs are reversible, thus suggesting that the subtype states are not permanent and underlie a certain degree of plasticity [32]. Targeting common epigenetic modifications, such as DNA methylation with DNA methyltransferase (DNTM) inhibitors or histone post-translational modifications with, e.g., HDAC inhibitors, is currently in the spotlight of epigenetic therapy. A comprehensive overview about recent advancements in the field of epigenetics in pancreatic cancer is provided, for example, by Paradise et al. [33]. To this date, however, none of the epigenetic therapies in PDAC has been successful. This is caused mainly by a lack of predictive biomarkers in these very heterogeneous tumors, which makes patient stratification almost impossible. In addition, the current generation of epigenetic drugs is simply not specific enough to combat the heterogeneity of PDAC.

#### 2.1.3. Metabolic Subtypes

Metabolic characterization of 38 pancreatic cancer cell lines by Daemen et al. revealed three subtypes with distinctive metabolic profiles: slow proliferative (in 34% of the cases), glycolytic (27%), and lipogenic (39%) [3]. The slow proliferative subtype showed a reduced proliferative capacity with a low level of carbohydrates and amino acids. The other two subtypes—glycolytic and lipogenic—demonstrated similar doubling times, yet had unique metabolic profiles. The glycolytic subtype displayed enrichment of glycolytic and serine pathways with reduced oxidative phosphorylation (OXPHOS), while the lipogenic subtype was enriched for various lipids as well as OXPHOS metabolites [3]. Interestingly, metabolic subtypes also revealed a strong correlation with transcriptomic subtypes. All cell lines with a glycolytic subtype presented association with the quasi-mesenchymal subtype, whereas most lipogenic cell lines overlapped with the classical subtype [3]. Of particular significance in this aspect is the identification of OXPHOS as an important target in cancer therapy. Consequently, cells with high OXPHOS dependency were susceptible to treatment with OXPHOS inhibitors metformin or oligomycin [34,35]. In line with this, various in vitro as well as in vivo studies suggested an anti-tumoral effect of metformin [36,37,38,39,40] (Table 1). However, addition of metformin to standard systemic therapy proved to be disappointing in advanced-stage pancreatic cancer [41,42]. More potent biguanide inhibitors, such as phenformin, could potentially be more effective in targeting mitochondrial metabolism. Phenformin demonstrates a more rapid entrance to the cell and mitochondria without transporters, resulting in an enhanced anti-neoplastic effect in OXPHOS high pancreatic cancer as well as other cancer entities [43,44,45,46,47,48] (Table 1). Consistent with these findings, Chuang et al. recently demonstrated a cell autonomous effect of phenformin on metastatic ability in murine lung cancer cells by specifically targeting dysfunctional mitochondria [49]. Other studies focused on inhibiting mitochondrial uptake of glutamine by targeting glutaminase 1 (GLS), which showed anti-proliferative effects in vitro but only minor effects on tumor growth in preclinical cancer models due to the metabolic adaptability of PDAC [50,51,52]. Moreover, the compound CPI-613 or devimistat that incorporates a dual inhibitory function by targeting α-ketoglutarate dehydrogenase (KGDH) and pyruvate dehydrogenase (PDH), two intermediates of the TCA-cycle, attained particular interest [53,54,55,56]. Recent findings encourage a potential clinical effectiveness of CPI-613 in treatment of metastatic PDAC, as the majority of patients treated with a combination of CPI-613 and mFOLFIRINOX demonstrated an objective response [57] (Table 1 and Table 2).

To date, targeting key metabolic factors has shown limited clinical impact, yet progress in the basic understanding of metabolic biology in pancreatic cancer is evident. Particularly, the identification of treatment-susceptible metabolic subtypes and stratification of metabolic biomarkers could serve as relevant tools to understand metabolic heterogeneity and refine the avenue of individualized therapy in pancreatic cancer.

#### 2.1.4. Immune-Landscape Heterogeneity

Based on the molecular subtypes in PDAC, it became obvious that these also contribute or even determine heterogeneity in the immune landscape found in pancreatic cancer. The presence of tumor-associated macrophages (TAM) and T-cells in the microenvironment or neoantigen levels in cancer cells can be very heterogeneous and determines response to (targeted) therapies and immune therapies. Consequently, there are three defined subtypes: the immune-escape phenotype, the immune-rich phenotype, and the mixed scenarios (reviewed by Karamitopoulou 2019) [65]. As implied by the name, the immune-escape phenotype is characterized by high levels of Tregs and M2 macrophages and low levels of effector T cells, which cause a poor host immune response and lead to an aggressive phenotype with poor prognosis [66]. The immune-rich phenotype on the other hand, is rich in effector CD4^+^ and CD8^+^ T cells and M1 macrophages and shows very low levels of Tregs and M2 macrophages. Tumors with this subtype display a high mutational frequency in genes involved in the intrinsic DNA damage response or the upregulation of the antigen presentation machinery and interferon signaling [67]. They are also more congruent with the classical subtype defined by Collison et al. or Moffit et al. [25,26]. Interestingly, a small subset of PDAC tumors that are arising mainly from IPMN (intrapapillary mucinous neoplasm) preneoplastic lesions display high microsatellite instability [68], mutations in mismatch repair (MMR), and are susceptible to immune checkpoint inhibitors [17]. To stimulate T-cell recruitment and tumor infiltration in the majority of immune-evasive PDAC, several strategies are being tested. One of these is the development and use of GVAX, which is a cancer vaccine composed of whole tumor cells that are genetically modified to secrete the immune-stimulatory cytokine GM-CSF (granulocyte-macrophage-colony stimulating factor). GVAX vaccination strongly correlates with an enhanced T-cell activity and corresponding tumor infiltration, enabling successful treatment with immune checkpoint inhibitors [69,70,71,72,73]. Others are examining the effects of live-attenuated Listeria monocytogenes bacterial strains (i.e., CRS-207) or CD-40 ligand therapy as tools to stimulate immune response in pancreatic cancer [58,74,75,76,77,78]. Currently, CAR (chimeric antigen receptor) T-cell technology is among the most intriguing immunotherapies. In PDAC, CAR-T cell therapy has been applied to target mesothelin, a tumor-associated antigen that is selectively expressed in malignant cells in various tumor entities [79]. Studies demonstrate that mesothelin is a promising candidate for a prognostic biomarker, with strong effects in vitro and in vivo [63,64]. A phase I clinical trial evaluated the activity of adoptive cell therapy by designing autologous mesothelin-specific CAR-T cells and testing them in a chemotherapy-resistant metastatic PDAC cohort, where it revealed strong anti-tumoral effects on liver metastasis in one patient with biopsy-proven mesothelin expression [80]. Additional promising targets are human epidermal growth factor receptor 2 (HER2), carcinoembryonic antigens such as CEACAM7, and PD-1/PD-L1, providing a novel and innovative option for CAR-T cell immunotherapy [59,60,61,62,81].

On the other hand, antibody-mediated blockade of PD-L1, which has led to remarkable success in many malignancies, has proven unsuccessful in pancreatic cancer [82,83]. A complex interplay of malignant cells and abundant tumor microenvironment (TME), strengthened by tumor heterogeneity aggravates successful therapy. Similar staggering results are delineated for CTLA-4 antagonists and even combinational efforts with standard chemotherapeutics proved disappointing so far [84,85]. The lack of efficacy seen with immune checkpoint inhibitors is conditional to the decisive processes of antigenicity and immunogenicity, as recently reviewed by Kabacaoglu and colleagues [86]. Consequently, novel approaches are being studied in order to overcome the highly immunosuppressive nature of PDAC tumor microenvironment. In this regard, Zhao and colleagues demonstrated an effective modulation of PDAC stroma by applying irreversible electroporation (IRE) to induce immunogenic cell death by membrane lysis or loss of hemostasis. By applying IRE in combination with anti-PD1 immune checkpoint blockade, Zhao et al. significantly suppressed tumor growth and prolonged survival in a murine orthotropic PDAC model [87]. Since both anti-PD1 and IRE are approved therapies, clinical trials investigating the effect of a combination of both approaches in patients are an attractive prospect to further enhance treatment efficacy in PDAC patients. In addition, other combination therapies together with checkpoint blockade might prove beneficial in the future.

#### 2.1.5. Inter-Tumor Heterogeneity between Tumor Sites

Within the same patient, PDAC primary tumor and metastasis sites are known to be rather homogenous regarding their gene expression, particularly of driver genes [26]. Analysis of sequencing data from various cancer types, including PDAC, revealed that within individual patients, the majority of driver gene mutations are shared between different tumor sites [88]. In PDAC, most genetic events, such as chromosomal rearrangements or genomic mutations occur rather early during cancer development, in the primary tumor itself [89,90]. That being said, metastatic sites do often possess different passenger or progression-associated gene mutations in comparison to the primary tumor [90]. Clonal populations that give rise to metastatic lesions are present within the primary tumor; however, these clones are genetically evolved from their original clone, which fits the clonal evolution model proposed by Peter Nowell [12,91]. Phylogenetic analyses suggested that metastatic lesions originate from different subclones within the primary tumor, implying significant evolution in order to achieve the optimal survival advantage [90,92]. Khoshchehreh et al. found that resetting epigenetic profiles of PDAC cells by using episomal vectors leads to reduced sphere formation in vitro and tumorigenicity in vivo [93]. Therefore, most likely it is epigenetic reprogramming, e.g., chromatin modifications, that promotes tumor progression as well as metastasis formation in PDAC [94].

Comparison of metastatic PDAC lesions seems to show that there is also significant heterogeneity between different metastatic sites: Peritoneal metastases (local dissemination) differ from liver and lung metastases (hematogenous dissemination). Even though the genomic landscape of those sites is almost identical, they may have distinct epigenetic programs, which cause differences in metabolism [94]. Comparative proteomic profiling of cells, isolated from different metastatic sites from one patient, showed that lung and liver metastases are more similar to each other than peritoneal metastasis [95].

#### 2.1.6. Inter-Tumor Heterogeneity in the Stroma

Stroma in PDAC is of utmost importance, as it makes up for most of the tumor’s volume and creates a very desmoplastic and hypoxic tumor microenvironment, which can promote proliferation, survival, and invasion of cancer cells [96]. Utilizing global transcriptomic profiling of stroma, Moffit et al. were able to define two types of stromata: “normal” and “activated” [26]. The “normal” stroma showed a relatively high expression of pancreatic stellate genes, while the “activated” stroma had a more inflammatory signature and was characterized by a diverse set of genes, including those associated with macrophages [26]. Moreover, patients with “activated” stroma typically have a worse prognosis and survival in comparison to patients with “normal” stroma. Indeed, multiple strategies were and are being tested to selectively target the heterogeneous stroma in PDAC, which are excellently reviewed by Jiang et al. [97]. So far, however, no stroma-targeting treatment has been successful in clinical trials. Future therapy approaches have to consider the stroma, the TME, and PDAC’s heterogeneity and improve the patient stratification process for potential trials. Moreover, the complex role of PDAC stroma is still mostly unclear. Further studying and understanding of its functions are very crucial in order to identify and validate potential therapeutic targets.

### 2.2. Intra-Tumor Heterogeneity

#### 2.2.1. Intra-Tumor Heterogeneity among Tumor Cells

Intra-tumor heterogeneity can be spatial—defined by the existence of multiple subclones within one tumor site, with distinctive driver or passenger mutations—or longitudinal—induced by therapies through clonal selection pressure over time, leading to or being cause by subclonal mutations [89,92,98].

So far, little is known about spatial intra-tumor heterogeneity in the epithelial compartment. Single-cell transcriptomic analyses of epithelial cells from different classical PDAC patients identified one cluster, which corresponded to a basal-like phenotype [99]. This suggests not only that basal-like cells might be more widespread, but also that intra-tumor heterogeneity in PDAC may be more complex than expected before. Sequencing of multiple primary tumor samples from different locations within a single tumor showed presence of geographically distinct subclones, each containing an independently expanded large number of cells [12]. In addition, single-cell RNA sequencing of primary PDAC tumors identified several subpopulations with different migratory and proliferative potentials within heterogeneous malignant ductal cells [98]. In mice carrying patient-derived xenografts, comparison of gene expression patterns between central and peripheral zones revealed significant differences [100]. Genes associated with motility and cytoskeleton were upregulated in the periphery, whereas genes involved in cell proliferation, transcription regulation, stress response, and carbohydrate metabolism were found in the center [100]. Since clonal development of cancer cells continues in metastases, the majority of lung and liver metastases are also characterized by polyclonality [101]. That being said, metastases are quite different in their mechanism and treatment sensitivity to the primary tumor [2,26].

Longitudinal intra-tumor heterogeneity plays a crucial role in developing resistance to cancer therapies [91]. Although cancer therapeutics, in particular chemotherapy, indeed eliminate most tumor cells, they also contribute to genomic instability in other cells. This creates therapeutic selection pressure, which favors the emergence of subclonal mutations, causing acquisition of resistance mutations and, therefore, a change in tumor phenotype [102]. This process is referred to as acquired resistance. In contrast, cells that are initially resistant possess the inherent ability to endure therapies. For therapy selection, it is important to recognize whether the detected acquired mutation is clonal, i.e., present in all the subclones, or not. Indeed, Seth et al. recently uncovered a multitude of functionally heterogeneous subpopulations of cells with differential degrees of drug sensitivity in pancreatic cancer, emphasizing the importance of understanding intra-tumor heterogeneity for successful therapy and targeting of resistant subclones [103].

#### 2.2.2. Intra-Tumor Heterogeneity in Stroma

Cancer-associated fibroblasts (CAFs) are key players within the stroma, as they produce components of extracellular matrix (ECM), which can then facilitate tumor growth and progression [96]. Elyada et al. were able to characterize the heterogeneity of PDAC stroma and define three different CAF subtypes by using single-cell RNA sequencing of human and mouse PDAC tumors: myofibroblastic, inflammatory, and antigen-presenting CAFs [96]. Myofibroblastic CAFs are typically found near the cancer cells and express high levels of smooth muscle actin (αSMA), while inflammatory CAFs are present in more desmoplastic areas of the tumor with low expression of αSMA [96]. The antigen-presenting CAFs, on the other hand, express MHC class II-related genes and induce T-cell receptor ligation [96]. Interestingly, CAF populations seem to be very dynamic: Their phenotype depends on their proximity to paracrine factors released by tumor cells or on their proximity to tumor cells themselves [104]. Using defined culture conditions, antigen-presenting CAFs can be converted into myofibroblastic CAFs [96]. Different CAF subtypes can have distinct effects on the tumor microenvironment and the immune response in PDAC. For example, genetic ablation of myofibroblastic CAFs resulted in more aggressive tumors with increased resistance to chemotherapy and stronger immune evasion as well as induction of EMT and stem-cell like phenotypes, highlighting the tumor-suppressive function of myofibroblastic CAFs [105]. Inflammatory CAFs, on the other hand, promote tumor progression by creating a tolerogenic microenvironment. That opens new avenues for therapy as inhibition of IL-6, produced by this CAF subtype, resulted in increased response rates to anti-PD-L1 immune checkpoint inhibitors in pancreatic tumors [106].

## 3. Novel Personalized Treatment Advancements in PDAC

Overall, it remains to be elucidated in which way targeting heterogeneity can be effectively translated in the clinic. The promise of future strategies to pursue individualized therapy relies on the precise selection of patients. Recent updates of the National Comprehensive Cancer Network (NCCN) in April 2021 endorse routine molecular subtyping and germline testing for pancreatic cancer patients [107]. This reasoning is based on a better understanding of the molecular basis of pancreatic tumors and positive findings in clinical trials. This described progress is evident in patients testing positive for mismatch repair deficiency (dMMR), who are—as in any cancer entities with this mutation—susceptible to the immune checkpoint inhibitor agent pembrolizumab, which demonstrated durable responses and has now been recommended as a second-line therapy for this PDAC subtype [17,108,109,110]. Another example are patients with a germline BRCA1/2 mutation that can be treated with the poly (adenosine diphosphate-ribose) polymerase (PARP) inhibitor olaparib, the treatment with which resulted in strong progression-free survival rates and, consequently, has been approved as maintenance therapy in metastatic PDAC [109,111]. In addition, high overall response rates and markedly improved overall survival were also achieved in the same patient subset by applying platinum-based chemotherapy regimens [112,113,114].

While there are many preclinical studies that suggest subtype stratification markers for more tailored therapy approaches (Table 1), ongoing clinical trials often do not implement them (Table 2). This is likely due to lack of feasible markers, missing marker validation, impracticable testing, or any combination thereof. However, the abovementioned examples demonstrate the need for both predictive as well as prognostic biomarkers and both are under intense investigation. While there are no reliable predictive markers available for PDAC, carbohydrate antigen 19-9 (CA19-9) is currently the only FDA-approved prognostic biomarker for clinical PDAC diagnosis. However, CA19-9 is also only partially reliable, as it has a sensitivity of only 50–75% and specificity of 83%, which can lead to misdiagnosis or false-positive and false-negative results (reviewed in [115]). To overcome these limitations, panels were investigated that include CA19-9 in combination with other markers to exceed the diagnostic value of CA19-9 alone [116,117,118,119]. In addition, non-coding RNAs, including microRNAs (miRNAs) and long non-coding RNAs (lncRNAs), showed prognostic value but the reproducibility of the results is controversial and the individual prognostic value of single non-coding RNAs is often too weak (reviewed in [115,120]). Consequently, panels consisting of multiple miRNAs or lncRNAs are under investigation [121,122].

Over the last decade, germline and somatic testing has been included in a clinical routine, accounting for inter-tumor heterogeneity on a genetic level. This lays the foundation for future targetable subtype specific alterations, which can be integrated into decision making for treatment of metastatic PDAC. Still, an expansion of these subtype-specific testings to metabolic, epigenetic, and transcriptomic levels and identification of reliable biomarkers for enhanced patient stratification are needed in order to further improve therapeutic strategies and patient survival and define the requirements for future preclinical investigations.

## 4. Studying Heterogeneity and Cell Fate

Initially, lineage tracing was used in classical developmental biology and subsequently applied in stem cell research [123,124,125,126,127]. In cancer research, lineage tracing is also increasingly shifting in focus, as intra- and inter-tumor heterogeneity, which can be attributed to the clonal evolution of the disease, are limiting factors for efficacious cancer therapy and contribute to poor prognosis [128,129,130]. In lineage tracing, target cells are uniquely labeled with heritable markers, which, in turn, allow cell lineages to be tracked through space and time. In the beginning, non-toxic dyes, which are, therefore, referred to as vital, were used to label and track cells and create “fate maps” based on this information. Initially, water-soluble dyes were used, which were later replaced by lipophilic dyes, as they can prevent diffusion to surrounding cells [131,132,133]. Following the discovery and cloning of the green fluorescent protein (GFP), fluorescent proteins were also widely to label cells [134,135]. The highest resolution is achieved with multicolor systems such as Brainbow [136,137]. These systems are based on several fluorescent markers, which are stochastically expressed and mediated by Cre recombinase and extensively utilized for neural fate mapping. Although dozens of color shades can be generated by combining different fluorescent proteins, the number of unique cellular identifiers is limited by the finite number of fluorescent reporters and their respective detection methods. Since the generation of the individual color code of the cells is based on Cre recombinase, the method can also only be used in transgenic organisms or cell systems derived thereof (reviewed in [138]).

### 4.1. Molecular Barcoding

Due to the need for a method that can individually label an increasing number of cells and driven by the ever-decreasing sequencing costs, molecular barcoding (also called cellular, DNA, or clonal barcoding) has emerged over the last 30 years as a versatile tool to study cellular heterogeneity, clonal lineages, and cell fate biomarker-free. In molecular barcoding, individual cells are labeled with unique and inheritable DNA sequences, so-called barcodes. Since the barcodes are stably integrated into the genome of the target cell, they also enable permanent tracking of cells, in contrast to earlier lineage tracing methods. A barcoding approach was first used by Walsh et al. in 1992 [139] to track neocortical rat cells and the term cellular barcoding was later coined by Schepers et al. [140]. The number of possible unique barcodes is basically unlimited and correlates with the length of the barcode sequence. Assuming a random arrangement of the four bases, there are 4^N^ unique combinations for a barcode sequence of length N. Thus, a six-nucleotide sequence yields 4096 unique barcodes, a nine-nucleotide sequence yields 262,144 and a 30-base pair barcode sequence is more than sufficient to theoretically uniquely mark every single cell in the human body [141]. In addition to the use of random barcode sequences [142], semi-random sequences can also be chosen, in which certain positions in the sequence are designated for specific bases (for instance, to ensure a balanced GC content [143]). Furthermore, barcodes can be generated enzymatically from an initially pristine sequence present in all cells, e.g., by shuffling specific sequences with recombinases [144,145] or by generating random mutations in the target sequence using CRISPR-Cas9 [146,147]. The traditional and most widely used approach to molecular barcoding is based on in vitro-generated barcode libraries, generally in the form of plasmids (exemplary shown in Figure 2A), which are then introduced into the target cells at a low copy number. Usually, viral vectors (mostly retroviruses [139] and lentiviruses [143]) are used, but other methods of gene transfer, such as microinjection of barcode constructs [148], are also possible. Most protocols aim for a low transduction efficiency to prevent cells from being labeled with multiple barcodes or from the same barcode being integrated into several different cells, since this could interfere with data quantification and potentially corrupt the lineage reconstruction [126,143]. From a methodological point of view, this approach is comparatively simple and based on established technologies. Moreover, the highest barcode diversities could thus far be achieved with in vitro-generated barcode libraries. For example, using a 30-nucleotide-long, semi-random DNA barcode, Bhang et al. [143] were able to create a barcode library with more than 10^7^ unique barcodes that could be used to track the fate of more than 1 million individual cancer cells. In addition, the in vitro-generated barcode libraries are characterized by high compactness, which results from the circumstance that the random arrangement of the four bases in even short barcodes generates tens of thousands of unique sequences. Another advantage resulting from the shortness of the sequences is that the barcodes can also be identified by short-read sequencing.

### 4.2. Applications of Molecular Barcoding

#### 4.2.1. Lineage Tracing and Fate Mapping

Once a barcode is introduced into a progenitor cell, it is then stably passed on to the progeny, rendering molecular barcoding a powerful tool to trace and reconstruct the development of cell lineages in a biomarker-free manner. As a result, barcoding has enabled the study of the clonal evolution of the hematopoietic system and helped to trace subclones that drive hematopoiesis after transplantation [127,149]. In addition, barcoding has been used extensively to elucidate the behavior of T cells; for example, to track the migration of T cells [140], the fate of naive T cells [150], and the kinship between T cell subpopulations [151].

Although the in vitro generation of barcodes is a simple technique and has so far achieved the highest diversity in available barcodes, it has some pitfalls. Barcoding of the target cells follows Poisson statistics, leaving a certain proportion of cells unlabeled. Furthermore, the applicability of such an approach is limited to biological questions and systems that are accessible for gene transfer. For example, in situ barcoding of hematopoietic stem cells is still not possible, which means that the cells must be labeled in vitro first and only then they can be transplanted into a recipient organism [144]. In order to overcome the limitations of in vitro-generated barcoding libraries and to label all cells in the biological system under investigation, the barcodes can also be derived from an initially pristine sequence, present in all cells. For this, a certain target sequence is continuously altered, either by shuffling the sequence with recombinases or by targeted introduction of mutations through Cas9. Thereby, different barcodes emerge from mutations and changes in the original sequence, which are also passed on to the progeny and serve as unique markers. However, this approach is technically more demanding and comes with a potentially lower barcode diversity.

Notably, Pei et al. have developed an artificial locus based on the Cre-loxP recombination system (called Polylox), which is used for in situ barcode generation, by applying a concept similar to Brainbow. The Polylox DNA cassette is composed of 10 loxP sites in alternating orientations, each separated by 178 base pairs. Expression of inducible Cre recombinase in specific cells or at certain time points induces shuffling of the Polylox sequence through excisions and inversions. This results in a multitude of unique combinations, which was used for barcoding of hematopoietic stem cells in vivo [144].

Similarly, the number of methods developed using the CRISPR/Cas9 system to generate barcodes that progressively evolve from a common origin sequence as cellular development constantly increases. GESTALT (genome **e**diting of **s**ynthetic **t**arget **a**rray for **l**ineage **t**racing) was the first reported approach that utilized evolving barcode sequences [147]. In GESTALT, barcodes consist of a series of continuously targeted CRISPR/Cas9 sites. The cells are identified by the unique combination of barcode marks (insertions and deletions caused by Cas9), which are also passed on to the progeny and, in turn, accumulate progressively more mutations. The authors of the study applied GESTALT for whole-organism lineage tracing in zebrafish and demonstrated that most cells in adult organs are derived from only a few embryonic precursor cells. Likewise, another Cas9-based barcoding method for single-cell, whole-organism lineage tracing was developed by Alemany et al. and coined the name ScarTrace [152]. ScarTrace is based on short insertions and deletions (“scars”) induced by Cas9 in multiple, targeted, histone-GFP transgene loci in zebrafish. Scarring can be induced by injecting the zygote either with Cas9 RNA or protein and a single-guide RNA (sgRNA) directed against GFP. The insertions and deletions caused at the targeted genomic site were then sequenced in order to reconstruct cell lineages. Using this approach, it could, for instance, be demonstrated that a small set of multipotent embryonic progenitors give rise to hematopoietic cells in the kidney marrow. In addition to that, fin regeneration in adult zebrafish could also be studied. In recent years, a number of other CRISPR/Cas9-based barcoding methods have been developed [146,153,154,155,156].

#### 4.2.2. Molecular Barcoding in Cancer Research

As previously described, tumor heterogeneity arises from mutations and clonal selection during cancer progression and, thus, is a hallmark of most cancers. Since it poses a significant challenge for diagnosis and clinical therapy, the investigation of tumor evolution has increasingly been shifting into focus [128,129,157,158]. While the past several decades of cancer research have led to a vastly increased understanding of cancer biology, the failure rate of novel oncology drugs in clinical use is still quite high. In this context, molecular barcoding can help to elucidate the causes of therapy failure and, thus, support the development of novel therapies (Figure 2B). Although it is known that therapy resistance is promoted by tumor heterogeneity, it is usually unclear whether the resistance is mainly mediated by pre-existing resistance mechanisms or whether it develops through a selective process throughout therapy [158,159,160,161].

To address this, Bhang et al. developed a highly complex barcode library that enabled high-resolution tracking of more than 1 million non-small cell lung cancer cells under drug treatment, called ClonTracer. In this study, they were able to show that resistance to the EGFR inhibitor erlotinib was inherent to a small, pre-existing subpopulation. Therefore, they concluded that combination therapies with different targets could challenge resistance and that ClonTracer could be used to optimize treatment regimens [143]. Similarly, barcoding was also used to investigate cancer cell populations in local recurrences after surgical removal of head and neck squamous cell carcinoma (HNSCC). Applying barcoding in a surgical in vivo model, Roh et al. observed a clonal substitution and massive reduction of clonal heterogeneity in local recurrence. Clones that were enriched in recurrences likely originated from ancestors shared with clones dominating in primary tumors. Those clones were found to be characterized by an epithelial-to-mesenchymal transition (EMT) program and displayed increased invasiveness. Consequently, the EMT-phenotype was targeted and the recurrence-free survival of tumor-bearing mice was significantly improved [162].

Another not yet fully understood process that contributes decisively not only to the lethality of cancer, but also to failure of therapies, is metastasis [163]. To study the molecular basis of metastasis that is driven by tumor heterogeneity and to probe the ability of subpopulations to contribute to various aspects of the disease, Wagenblast et al. have developed a barcoding-based mouse model of breast cancer heterogeneity [164]. To this end, mammary carcinoma cells were labeled with a molecular barcode via retroviral infection and injected orthotopically into immunocompromised recipient mice. After metastatic spread to brachial lymph nodes, blood, lungs, livers, and brains, tumor cells were harvested and barcode populations within each tissue were quantified. As a result, it was possible to demonstrate that distinct populations contributed to lymph node and hematogenous metastases. Moreover, a clear overlap between abundant clones in the hematogenous metastases and in circulating tumor cells was discovered. It was also possible to reconstruct the evolution of circulating tumor cells from underrepresented subpopulations in the primary tumor. From these cells, only a subset had the additional capacity to colonize secondary sites. It could be further explained that the emergence of circulating tumor cells and their ability to form metastases was directly linked to vascular mimicry (a process in which tumor cells form tubular structures for blood and nutrient transportation independent of classical angiogenesis) driven by increased expression of two secreted proteins: Serpine2 and Slpi. In addition, they elucidated that Serpine2 and Slpi act as anticoagulants to sustain perfusion. By these means, Serpine2 and Slpi promote tumors by enhancing the blood supply and, at the same time, providing new opportunities to metastasize [164].

Similarly, molecular barcoding in combination with unbiased genomic analysis and a small-scale in vivo functional screen was used by Chuang et al. in order to identify pharmaceutically susceptible targets for the inhibition of metastatic ability [165]. Tumor formation and barcoding were initiated in a mouse model of human lung adenocarcinoma (Kras^LSL-G12D/+^;Trp53^flox/flox^ mice that also contained a Rosa26^LSL-tdTomato^) with a pool of barcoded lentiviral vectors that expressed Cre recombinase. After tumor formation, cancer cells were isolated from individual primary tumors as well as metastases from multiple sites. Barcoding then allowed distinguishing between nonmetastatic primary tumors (T_nonMet_) and primary tumors that give rise to macrometastases (T_Met_). Subsequently, RNA sequencing-based gene expression profiling, performed on T_nonMet_ primary tumors, T_Met_ primary tumors, and macrometastases revealed distinct molecular states. Based on this, a pharmaceutically targetable pro-metastatic CD109-Jak-Stat3 axis was identified through an in vivo, small-scale screening and molecular analyses.

To shed further light on whether drug resistance is inherent to subpopulations of cancer cells or acquired through selection under therapy, Umkehrer et al. developed [156] CaTCH, a technique combining sophisticated clone tracing with the ability to isolate specific clones. The isolation of such clones is not only possible at any time point, but also from complex cell populations. In CaTCH, cells are first transduced with a dCas9 construct and subsequently labeled with DNA barcodes that are fused with an inducible (activatable by barcode-specific sgRNAs) green fluorescent protein (GFP) reporter. After experimental selection under therapy, barcode representation is identified by next-generation sequencing. This enables the design of single-guide RNAs that are complementary to the clone of interest. Afterwards, the sgRNA constructs are transduced into the heterogeneous cell population, specifically activating GFP expression in the desired clone. This enables isolation of the clone of interest by fluorescence-activated cell sorting. Furthermore, if cells are preserved prior to the selection process, specific clones can be isolated from the pool, which will ultimately give rise to resistant clones. Consequently, the isolation of the clonal pair of cells is still possible, no matter if the clonal pair of cells is in the treatment-naive, intermediate, or therapy-resistant state. By comparing the therapy response of the cells acquired at different stages of resistance, it can be investigated if drug tolerance was inherent or acquired. By applying CaTCH in an in vivo melanoma mouse model, Umkehrer et al. could show that RAFi/MEKi resistance is mostly acquired and that the majority of clones are capable of achieving this state.

#### 4.2.3. High Throughput Screens

Molecular barcoding can not only be used to identify new therapeutic targets by identifying pathophysiological processes, but also by being applied in advanced drug screenings [166]. Large-scale screening of drugs is typically only performed in vitro and, although one-by-one drug screening of large drug libraries in vivo is potentially possible, it is also labor intensive, often costly, and ethically problematic. To overcome these limitations and enable the analysis of numerous compounds in parallel in the same mouse, we developed a multiplexed, small-molecule, in vivo screening platform. This platform utilizes molecular barcoding to screen for modulators of metastatic seeding in PDAC [142]. Due to barcoding, a multitude of drugs can be screened simultaneously while reducing experimental time, costs, and resources. Pooled screening is enabled by labeling the same (polyclonal) cells with individual barcodes to generate 96 uniquely barcoded cell variants of a metastatic pancreatic cancer cell line so that individual variants can be tracked in a cell pool. Each cell variant can then be pretreated with an individual compound in vitro. Following the treatment, the treated cells are pooled and injected into a recipient mouse. The effect of each drug on metastatic seeding ability can then be analyzed by determining the barcode representation in postseeding samples in relation to the preinjection samples with Next Generation Sequencing (NGS). Using this platform to screen over 700 compounds and 300 internal controls, the lipase ABHD6 was identified as a novel regulator of cancer cell adhesion [142]. Instead of using thousands of mice, this experimental setup required only 36 mice, thus demonstrating the power of using barcodes as a screening approach. Barcoding can not only reduce the technical variability that is inherently linked to in vivo assays but it also identifies novel regulators of metastatic ability that are unlikely to be identified with conventional approaches.

## 5. Conclusions

Tumor heterogeneity and clonal evolution during cancer progression as well as under therapy are major obstacles for improved treatment options in advanced stage PDAC. DNA barcoding technologies provide a powerful tool to identify targetable (sub-)populations and resistant cell clones. This method will most likely be one of the keys to advanced personalized treatment strategies in this devastating cancer. With many novel and exciting developments in the field of single-cell analysis, it will be interesting to see how these strategies will be deployed alongside conventional methods to further investigate cancer heterogeneity in the future.

## Figures and Tables

**Figure 1 cancers-13-04187-f001:**
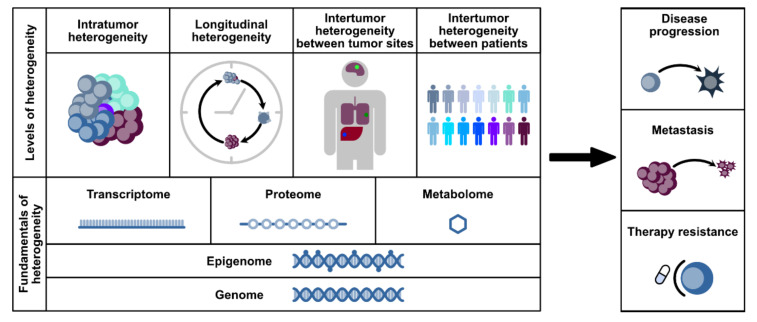
Levels of heterogeneity. Heterogeneity in tumors relies on different mechanisms within the genome, epigenome, transcriptome, proteome, or metabolome. Based on that, there can be distinct levels of heterogeneity: intra-tumor, longitudinal, and inter-tumor heterogeneity between tumor sites and between patients. Studying heterogeneity is imperative due to its crucial role in disease progression, metastasis, and therapy resistance.

**Figure 2 cancers-13-04187-f002:**
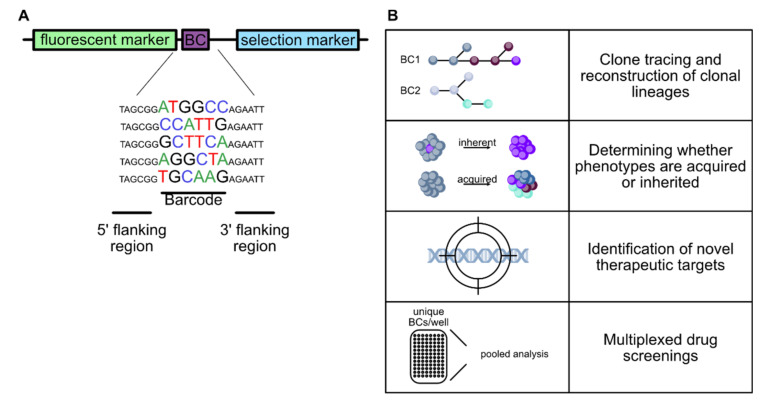
Molecular barcoding in cancer research. (**A**) Schematic of a barcoding construct exemplary with a short, six- nucleotide, random barcode sequence (BC). Often, fluorescent proteins such as GFP are also part of the construct to facilitate the tracking of barcoded cells. In addition, selection markers are often used to exclude unlabeled cells. (**B**) Overview of the applications of molecular barcoding in cancer research. Barcodes can serve as heritable tags that allow biomarker-free clone tracking and cellular lineage reconstruction. In addition, they can be used to determine whether certain phenotypes or characteristics in a population, e.g., therapy resistance or metastatic ability, are acquired or inherent. Furthermore, barcodes can be used to identify new molecular drug targets and simplify multiplexed drug screenings.

**Table 1 cancers-13-04187-t001:** Novel therapeutic targets discovered in preclinical trials considering tumor heterogeneity for patient stratification.

Compound of Interest	Putative Molecular Target	Subtype Specificity	Pubmed-ID & Year of Publication
Metformin	HNF4G via AMPK COX6B2	SMAD4-deficiency High levels of COX6B2	32737864; 2021 [36] 32415061; 2020 [38]
Phenformin	Mitochondrial Complex I	High OXPHOS	33294863; 2020 [45]
CB-839+-ß-lapachone	GLS 1/NQO1	Mutant KRAS + NQO1 overexpression	26462257; 2015 [50]
CD40 ligand	CD40	Immune-poor/high levels of M2 macrophages	27906162; 2017 [58]
Chimeric antigen receptor-engineered T cells	Human epidermal growth factor receptor 2 (HER2) CEACAM7 (Carcinoembryonic antigen-related cell adhesion molecule 7) Immune Checkpoint Inhibitors PD-1/PD-L1 Mesothelin	HER-2 positive tumors Exclusive expression in PDAC tumors PD-1/PD-L1 overexpression Mesothelin high PDAC	33742099; 2021 [59] 30121627; 2019 [60] 33479048; 2021 [61] 32637575; 2020 [62] 29859625; 2018 [63] 28929447; 2017 [64]

**Table 2 cancers-13-04187-t002:** Current ongoing clinical trials evaluating the efficacy of promising compounds for treatment of pancreatic cancer.

Compound	Treatment Regime	Subtype Specific Patient Stratification	Clinical Phase	NCT-Number
CPI-613 (Devimistat)	FOLFIRINOX	No	Phase 1/2 Recruiting	NCT03699319
Mitazalimab (CD40 agonist)	mFOLFIRINOX	No	Phase1b/2 Not yet recruiting	NCT04888312
LOAd703 (oncolytic adenovirus encoding TMZ-CD40L)	Gemcitabine+nab-paclitaxel+/-atezolizumab (anti PD-L1)	No	Phase 1/2a Recruiting	NCT02705196
CART-meso cells	/	No	Not applicable	NCT03638193
	/	No	Phase 1	NCT03323944
Anti-CEA CAR-T cells	Systemic chemotherapy regimens	CEA-expressing PDAC with livermet	Phase 2b Recruiting	NCT04037241
	/	CEA expressing tumors	Phase1/2 Recruiting	NCT04348643
GVAX	Nivolumab, Ipilimumab, Cyclophosphamide, CRS-207	No	Phase 2 Recruiting	NCT03190265
	Epacadostat, Pembrolizumab, CRS-207, Cyclophosphamide	No	Phase 2 Recruiting	NCT03006302

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
