# Peer review of "Unraveling Tumor Heterogeneity by Using DNA Barcoding Technologies to Develop Personalized Treatment Strategies in Advanced-Stage PDAC"

_cancers, 2021, doi:10.3390/cancers13164187_

Round 1

Reviewer 1 Report

This article reviews the potential applications of molecular barcoding technologies towards studying multi-level heterogeneity in PDAC. Because tumor heterogeneity plays a central role in understanding and treating cancers, such applications should eventually contribute to precision cancer medicine.

The manuscript is thoroughly designed, well structured, and clearly written. It starts from describing the overall context of multi-level heterogeneity in pancreatic cancer and its central role in research and therapy development. It then dedicates the second half introducing the basic methodologies and various applications of molecular barcoding technologies. Large volume of references are integrated and well summarized. The schematic plots are especially well designed and will be very helpful for readers to grasp the big picture of this subject.

In general, I found no major issues that hinder the publication of this manuscript. A few minor issues are specified below.

Page 1, Line 3-4 “Among the main reasons that make PDAC one of the hardest-to-treat cancers is its multi-level heterogene-ity, ...“ It is beneficial to briefly list other “main reasons” that render PDAC a tough cancer to treat, before highlighting tumor heterogeneity.

Page 2, Line 19-20 “Tumor heterogeneity is defined by phenotypical and morphological differences between tumor cells, …” Although most types of heterogeneity discussed in this manuscript are those “between tumor cells” or at the cell level, in general, tumor heterogeneity is also prominent at the tissue/region level (between microenvironments/subregions), which is of particular interest in digital pathology.

Page 7 vs. Page 9. Two sections are numbered “3”. Please correct this.

Page 9, Line 5 “… to the clonal evolution of the disease, are are limiting factors for efficacious cancer” Obviously a typo. Please correct.

Author Response

Reviewer 1:

This article reviews the potential applications of molecular barcoding technologies towards studying multi-level heterogeneity in PDAC. Because tumor heterogeneity plays a central role in understanding and treating cancers, such applications should eventually contribute to precision cancer medicine.

The manuscript is thoroughly designed, well structured, and clearly written. It starts from describing the overall context of multi-level heterogeneity in pancreatic cancer and its central role in research and therapy development. It then dedicates the second half introducing the basic methodologies and various applications of molecular barcoding technologies. Large volume of references are integrated and well summarized. The schematic plots are especially well designed and will be very helpful for readers to grasp the big picture of this subject.

In general, I found no major issues that hinder the publication of this manuscript. A few minor issues are specified below.

Response to Reviewer 1:

We highly appreciate that the Reviewer finds our manuscript appealing and thank the Reviewer for their kind comments. We have addressed the issues pointed out by the Reviewer as detailed below.

Q1: Page 1, Line 3-4 “Among the main reasons that make PDAC one of the hardest-to-treat cancers is its multi-level heterogeneity, ...“ It is beneficial to briefly list other “main reasons” that render PDAC a tough cancer to treat, before highlighting tumor heterogeneity.

A1: The suggested additions to page 1 line 3-4 have been included and will certainly prove helpful to readers.

Q2: Page 2, Line 19-20 “Tumor heterogeneity is defined by phenotypical and morphological differences between tumor cells, …” Although most types of heterogeneity discussed in this manuscript are those “between tumor cells” or at the cell level, in general, tumor heterogeneity is also prominent at the tissue/region level (between microenvironments/subregions), which is of particular interest in digital pathology.

A2: We appreciate the comment made by the reviewer and have complemented the manuscript accordingly.

Q3: Page 7 vs. Page 9. Two sections are numbered “3”. Please correct this.

A3: The mislabeling has been corrected.

Q4: Page 9, Line 5 “… to the clonal evolution of the disease, are are limiting factors for efficacious cancer” Obviously a typo. Please correct

A4: Thank you for bringing the typo to our attention. Corrections were made.

Reviewer 2 Report

The manuscript is well written, it concerns the most recent knowledge on detection of tumour heterogeneity in PDCA and its implications for therapy.

Data are described in a comprehensive manner and innovative technologies based on DNA barcoding for heterogeneous cell identification are discussed.

I suggest authors to present in more details data summarized in tables 1 and 2, which are of major interest for the topic, for instance explaining the diverse subtypes of tumour analyzed (which differ with respect to data in the above paragraphs of the manuscript) and, at least, the state of progression of clinical trials.

Overall, I recommend the manuscript for the publication in Cancers.

Author Response

Reviewer 2:

The manuscript is well written, it concerns the most recent knowledge on detection of tumour heterogeneity in PDCA and its implications for therapy.

Data are described in a comprehensive manner and innovative technologies based on DNA barcoding for heterogeneous cell identification are discussed.

I suggest authors to present in more details data summarized in tables 1 and 2, which are of major interest for the topic, for instance explaining the diverse subtypes of tumour analyzed (which differ with respect to data in the above paragraphs of the manuscript) and, at least, the state of progression of clinical trials.

Overall, I recommend the manuscript for the publication in Cancers.

Response to Reviewer 2:

We thank the Reviewer for their constructive comments to our manuscript and appreciate their suggestions. In table 1 we have listed interesting data from preclinical studies that suggest a potential patient stratification according to subtype-specific markers. We considered this data promising to mention, as it emphasizes the need for practicable biomarker stratification in trials and treatment strategies. In table 2 we have listed examples of current clinical trials that test new treatment strategies. Here, we have included now the state of progression of the trial as additional information. The discrepancy in both tables emphasizes the fact that even though subtype-specific markers are discovered in preclinical studies they are often not translated into trial design and subsequently into clinical practice. We do hope that the provided information is now more clear and strengthens the high importance of the topic.

Reviewer 3 Report

The authors present a review article, describing the role of modern techniques to identify tumor heterogeneity in pancreatic cancer aiming to increase personalized treatment for patients with advanced pancreatic cancer. The review is well-written and interesting for the readers of Cancers. I have only some minor suggestions to improve the paper.

  • Besides immunotherapy as stand-alone treatment for pancreatic cancer, or in addition to modern chemotherapeutic regimen, mainly in vitro studies describe on promising outcomes of combining immunotherapy for pancreatic cancer with local ablative treatment, for instance irreversible electroporation (Zhao et al. Nature Communications 2019). The authors could elaborate on possible future combination treatments to further enhance treatment efficacy.
  • Authors describe only briefly the lack of predictive biomarkers for pancreatic cancer. It would be interesting for the readers if the authors could add a paragraph describing the use of known biomarkers for pancreatic cancer (i.e. CA19.9, miRNA etc.) and its limitations for clinical practice (i.e. high prognostic but low predictive value). 
  • Despite evidence suggesting a possible survival benefit of treating patients with pancreatic cancer with BRCA mutations with PARP-inhibitors, these patients may also benefit from platinum-based chemotherapy regimen (Blair et al. JACS 2018, Wattenberg Br J Canc 2020).

Author Response

Reviewer 3:

The authors present a review article, describing the role of modern techniques to identify tumor heterogeneity in pancreatic cancer aiming to increase personalized treatment for patients with advanced pancreatic cancer. The review is well-written and interesting for the readers of Cancers. I have only some minor suggestions to improve the paper.

Response to Reviewer 3:

We appreciate that the Reviewer finds our manuscript interesting and useful for the readership of Cancers. We thank the Reviewer for their suggestions and constructive improvements that are addressed in detail below.

Q1: Besides immunotherapy as stand-alone treatment for pancreatic cancer, or in addition to modern chemotherapeutic regimen, mainly in vitro studies describe on promising outcomes of combining immunotherapy for pancreatic cancer with local ablative treatment, for instance irreversible electroporation (Zhao et al. Nature Communications 2019). The authors could elaborate on possible future combination treatments to further enhance treatment efficacy.

A1: We thank the Reviewer for bringing the combination of immune checkpoint inhibitors with local ablative treatment to our attention. A corresponding paragraph was added to the manuscript and will certainly be of interest to the readership.

Q2: Authors describe only briefly the lack of predictive biomarkers for pancreatic cancer. It would be interesting for the readers if the authors could add a paragraph describing the use of known biomarkers for pancreatic cancer (i.e. CA19.9, miRNA etc.) and its limitations for clinical practice (i.e. high prognostic but low predictive value). 

A2: We agree with the Reviewer that biomarkers are of great importance and complemented the paragraph with additional information accordingly.

Q3: Despite evidence suggesting a possible survival benefit of treating patients with pancreatic cancer with BRCA mutations with PARP-inhibitors, these patients may also benefit from platinum-based chemotherapy regimen (Blair et al. JACS 2018, Wattenberg Br J Canc 2020).

A3: We thank the Reviewer for their suggestion. A corresponding addition was made in the manuscript.